# Cardiovascular Complications of COVID-19 among Pregnant Women and Their Fetuses: A Systematic Review

**DOI:** 10.3390/jcm11206194

**Published:** 2022-10-20

**Authors:** Shirin Yaghoobpoor, Mobina Fathi, Kimia Vakili, Zohreh Tutunchian, Mina Dehghani, Ashkan Bahrami, Ramtin Hajibeygi, Samira Eslami, Tina Yaghoobpour, Mohammadreza Hajiesmaeili

**Affiliations:** 1Student Research Committee, Faculty of Medicine, Shahid Beheshti University of Medical Sciences, Tehran, Iran; 2School of Medicine, Isfahan University of Medical Sciences, Isfahan, Iran; 3Faculty of Medicine, Kashan University of Medical Science, Kashan, Iran; 4Firoozgar Hospital, Iran University of Medical Sciences, Tehran, Iran; 5Department of Clinical Sciences, School of Veterinary Medicine, Shiraz University, Shiraz, Iran; 6Anesthesia and Critical Care Department, Critical Care Quality Improvement Research Center, Loghman Hakim Hospital, Shahid Beheshti University of Medical Sciences, Tehran, Iran

**Keywords:** COVID-19, SARS-CoV-2, pregnancy, cardiovascular complications, pre-eclampsia

## Abstract

Background: COVID-19 is a viral infectious disease leading to a spectrum of clinical complications, especially cardiovascular. Evidence shows that this infection can potentially accompany a worse outcome in pregnant women. Cardiovascular complications in mothers and their fetuses are reported by previous studies. Objective: In this systematic review, we aim to investigate the cardiovascular complications of COVID-19 during pregnancy in the mothers and fetus, according to the published literature. Method: We systematically searched the online databases of PubMed, Scopus, Web of Science, and Google Scholar, using relevant keywords up to April 2022. We included all observational studies reporting cardiovascular complications among COVID-19-affected pregnant women and their fetuses. Results: We included 74 studies containing 47582 pregnant COVID-19 cases. Pre-eclampsia, hypertensive disorders, cardiomyopathy, heart failure, myocardial infarction, thrombosis formation, alterations in maternal–fetal Doppler patterns, and maternal and fetal arrhythmia were reported as cardiovascular complications. The highest incidences of pre-eclampsia/eclampsia among COVID-19 pregnant cases, reported by studies, were 69% and 62%, and the lowest were 0.5% and 3%. The highest and lowest incidences of fetal bradycardia were 20% and 3%, and regarding fetal tachycardia, 5.4% and 1%, respectively. Conclusion: SARS-CoV-2 infection during pregnancy can potentially be associated with cardiovascular complications in the mother, particularly pre-eclampsia and heart failure. Moreover, SARS-CoV-2 infection during pregnancy can potentially cause cardiovascular complications in the fetus, particularly arrhythmia.

## 1. Introduction

Since 2020, the COVID-19 pandemic has caused considerable changes in the clinical management of pregnant women, their fetuses or neonates, and healthcare delivery, as the accessibility of healthcare resources, scientific data, and infection rates continue to evolve. COVID-19 is a highly transmittable virus with various symptom presentations from mild or asymptomatic to critical conditions [1]. Its mild symptoms include dry cough, gastrointestinal symptoms, anosmia, dyspnea, sore throat, headache, ageusia, and fatigue. Moreover, serious issues including thromboembolism and cardiovascular complications such as acute myocardial damage, cardiogenic shock, and arrhythmia, have been indicated in adult COVID-19 patients [2,3]. The prevalence of COVID-19 infection across pregnant women has been announced to be 14–15%, with most (50–90%) pregnant women being asymptomatic [4,5,6,7]. Only a small portion of pregnant women show severe symptoms, principally in the third trimester of pregnancy. Among those with severe symptoms, the risk of death and severe complications is higher [8,9,10,11].

It was believed that because of the weakened immune system, which is noticeable by certain raised pro-inflammatory cytokines and reduced lymphocytes in pregnant women, they are in a riskier situation in the case of encountering COVID-19 [12], but in recent studies, it was described that pregnancy does not raise the risk of COVID-19 infection; however, it aggravates the clinical outcome of COVID-19 in contrast to non-pregnant women [13,14].

In addition to the common symptoms, cardiovascular symptoms and complications such as hypoxemia and heart failure due to peripartum cardiomyopathy (PPCM) were well described [15,16]. Moreover, in critical and severe COVID-19 pregnant cases, myocardial injury and bradycardia are especially noted [17]. COVID-19 infection in pregnant women can result in an elevated risk of pregnancy complications such as pre-term premature rupture of membranes (PPROM), pre-term birth, pre-eclampsia, and cesarean. It may cause maternal death in rare cases.

The primary aim of the current systematic review study is to summarize the current literature on the maternal and fetal cardiovascular outcomes of COVID-19 infection reported during pregnancy. Moreover, we aim to investigate the potential association between COVID-19 infection during pregnancy and cardiovascular outcomes including pre-eclampsia and hypertensive disorders.

## 2. Methods

This study was performed based on the Preferred Reporting Items for Systematic Reviews and Meta-analysis (PRISMA) protocol for reporting systematic reviews.

### 2.1. Search Strategy

This article assessed qualitative data on the COVID-19 cardiovascular complications in pregnant women. Two authors (A.B. and R.H) searched the medical subject headings (MeSH) in the PubMed, Scopus, Web of Science, and Google Scholar databases to find all related research articles. All relevant articles published before April 2022, mostly in 2020–2021, were evaluated. The following search terms were used ((“Pregnant Women”[Mesh]) OR (“fetus”[Mesh]) OR (“Pregnancy”[Mesh])) AND (“COVID-19”[Mesh]) AND ((“Heart”[Mesh]) OR (“Cardiovascular Abnormalities”[Mesh]) OR (“Cardiovascular System”[Mesh]) OR (cardio*)). In addition, the reference list of the relevant papers was reviewed to avoid missing any publication. All searched studies were included in the Endnote software for screening.

### 2.2. Inclusion and Exclusion Criteria

We included studies with the following criteria: (1) observational studies with prospective, case-control, cross-sectional, case series, or case report designs; (2) studies that reported the cardiovascular complications of COVID-19 among pregnant women; (3) studies that reported the cardiovascular complications of COVID-19 among the fetuses of affected pregnant women; (4) English language studies; and (5) studies that diagnosed COVID-19 by polymerase chain reaction (PCR) during pregnancy.

We excluded non-English articles, interventional studies, review articles, studies that did not use PCR to diagnose SARS-CoV-2 infection, case reports that diagnosed COVID-19 after delivery, and studies that did not report any cardiovascular disease complications among affected pregnant women or their fetuses.

Two authors (M.D. and R.H.) conducted title–abstract and full-text screening independently and in parallel. The disagreements between authors were finally resolved via consultation with a third author (S.Y). The title, abstract, and full-text screening of all the related articles resulted in 18 final studies. The extracted information from each of the identified studies is as follows: sample size, the number of pregnant women with a positive COVID-19 test, the number of pregnant women with a negative COVID-19 test, trimester, the mean age of positive and negative subjects, and key findings.

### 2.3. Data Extraction

Two authors (M.D. and A.B.) extracted the following information: first author’s name, the date of publication, the country of origin, the demographic characteristics of the participants (age range or mean age and gestational age), the number of pregnant women and number of COVID-19-affected pregnant cases, and cardiovascular complications reported in the COVID-19-affected pregnant women and their fetuses.

### 2.4. Quality Assessment

Two authors (M.D. and A.B.) conducted a critical appraisal of all observational studies to assess the quality of those that were included. Disagreements were resolved through consensus with another author (S.E.). We separately applied the Newcastle–Ottawa scale (NOS) checklists for cohort and case-control studies. We used the Joanna Briggs Institute (JBI) checklist to assess case series and case reports (Appendix A).

This study was approved by the Iranian National Committee for Ethics in Biomedical Sciences (Code of Ethics: IR.SBMU.RETECH.REC.1400.840).

## 3. Results

### 3.1. Study Selection

A total number of 1386 records were identified by database searching. After removing duplicated articles, authors screened the titles and abstracts, and 214 articles remained. In the next step, 74 articles were included in our systematic review study after considering the eligibility of full texts (Figure 1).

### 3.2. Demographic Characteristics and Cardiovascular Comorbidities

In total, 41 cohorts, cross-sectional, or case-control studies were included, containing a total of 47,582 pregnant COVID-19 patients with a mean age of 27.8 years. In addition, 38 case reports and case series studies were reviewed. Hypertension was a cardiovascular comorbidity reported by several studies. The highest prevalence of this comorbidity in pregnant cases was 51.2% [18] and 33.4% [19], and the lowest was 1.5% [20] and 1.4% [21] (Table 1, Table 2, Table 3 and Table 4).

### 3.3. Pre-Eclampsia and Hypertensive Disorders

Hypertensive disorders during pregnancy, including pre-eclampsia and eclampsia, were among the reported cardiovascular events in pregnant COVID-19 cases. Pre-eclampsia is a condition diagnosed by a blood pressure of >140/90 mmHg coincidence with proteinuria or end-organ dysfunction after the 20th week of pregnancy [22]. In a group of studies, pre-eclampsia and hypertension were reported as complications that occur in pregnant women infected with SARS-CoV-2. Adhikari et al., in a cohort study, evaluated the adverse outcomes of COVID-19 on 245 pregnant patients. Wu et al. also reported that among 29 pregnant COVID-19 cases, two had hypertensive disorders (6.90%) [23]. Vaezi et al. reported severe pre-eclampsia in three and hypertension in one of 24 COVID-19-positive pregnant patients [24]. Leal et al. indicated that hypertension and other cardiovascular diseases occur in 10.3% of pregnant COVID-19 cases as a comorbidity [25].

In some of the studies reviewed, two groups of pregnant women with and without COVID-19 are compared, and pre-eclampsia is considered to occur in COVID-19-infected pregnant women more than in non-infected controls. Pre-eclampsia with severe features was a cardiovascular outcome observed among 11% of recorded patients, which was higher than the healthy controls (*p* < 0.05 [26]. Jayaram et al. [27] studied 75 pregnant women with coronavirus disease in a cohort study. Of the cases, 24% were diagnosed with pre-eclampsia, which was higher than the controls (*p* < 0.001 Abedzadeh-Kalahroudi et al. [28] conducted a cohort study on 150 pregnant women (56 cases confirmed with positive COVID-19 and 94 controls). Pre-eclampsia occurred in 19.8% of cases and 7.4% of controls (*p* = 0.037). In a study by Ahlberg et al. [29] on 155 COVID-19-positive and 604 matched negative pregnant women, pre-eclampsia was seen in 7.7% of positive women and 4.3% of negative controls. Antoun et al. [30] studied a group of 23 pregnant patients prospectively, 16 of whom were confirmed with positive COVID-19 in the third trimester. A total of two out of 16 cases (10.5%) developed pre-eclampsia. Cruz Melguizo et al. [20] compared a group of 1347 COVID-19-infected with 1607 non-infected pregnant women. They reported that a higher proportion of SARS-CoV-2-infected women were diagnosed with severe pre-eclampsia (40.6% vs. 15.6%; *p* < 0.001), whereas moderate pre-eclampsia occurred more frequently in the non-infected group. A higher rate of pre-eclampsia/eclampsia was observed in pregnant patients infected with COVID-19 compared to healthy pregnant women, as demonstrated by Epelboin et al. (4.8% vs. 2.2%, *p* < 0.001) [31]. Gurol-Urganci and colleagues [32] showed a higher risk of pre-eclampsia in SARS-CoV-2-infected patients in their study (3.9%; *p* < 0.001). Mahajan et al. [33] evaluated the effect of COVID-19 on multiple gestation pregnancies. Based on their study, the risk of pre-eclampsia/eclampsia was higher in MGPs with COVID-19 than MGPs before the pandemic (50.0% vs. 12.7%) and singleton pregnancies (41.6% vs. 7.9%). Papageorghiou et al. [34] indicated a risk ratio of 1.86 for pre-eclampsia among pregnant COVID-19 patients. A prospective cohort study on 199 women (66 COVID-19-infected and 133 non-infected) indicated an adjusted risk ratio of 2.02 for pre-eclampsia [35]. Serrano et al. [36] indicated a higher risk for developing pre-eclampsia in 231 pregnant women in comparison with the general population (19.0% vs. 13.1%; *p* = 0.012). Soto-Torres et al. reported no significant difference in the abundance of hypertensive disorders between SARS-CoV-2 positive and negative pregnant cases (16% vs. 17.5%; *p* = 0.8); however, they recorded a trend toward greater rates of pre-eclampsia in the positive cases (19.8% vs. 10.7%; *p* = 0.08) [10]. Jering et al. studied 6380 pregnant women with coronavirus disease and 400,066 controls without COVID-19. They reported that the incidence rate of pre-eclampsia was slightly higher among SARS-CoV-2 infected pregnant cases than pregnant cases without this infection (8.8% vs. 6.8%; *p* < 0.001). In addition, they found that eclampsia among pregnant COVID-19 patients was correlated with greater odds of in-hospital death or mechanical ventilation [37].

Some studies concentrated on the severity of COVID-19 and its effect on the incidence of pre-eclampsia. Arslan et al. [38] observed pre-eclampsia in 11 out of 110 pregnant patients with COVID-19, with a higher percentage in non-severe patients (*p*-value = 0.09). A prospective observational study, conducted by Mendoza et al. [39], studied 34 cases of non-severe and eight cases of severe COVID-19 pregnant women. Five cases, all infected by severe COVID-19, developed pre-eclampsia. Osaikhuwnomwan et al. [40] described pre-eclampsia as the most common co-morbidity of pregnancy. They demonstrated that pre-eclampsia was highly related to severe COVID-19 (*p* = 0.028).

Several studies reported no important difference in the incidence or risk of pre-eclampsia between COVID-19 infected and non-infected pregnant women. Rosenbloom et al. [41] reported hypertensive disorders of pregnancy among pregnant patients, including gestational hypertension and pre-eclampsia. The prevalence of hypertensive disorders of pregnancy among cases was not significantly higher than that of controls (28.9% vs. 27.7%; *p* = 0.84). In the case-control study conducted by Daclin et al. [42], there was no significant difference in the incidence of pre-eclampsia between pregnant women before the pandemic and pregnant women infected by COVID-19 (*p*-value = 0.36). Ferrara et al. [43] demonstrated no significant difference in the incidence of pre-eclampsia/eclampsia between pregnant women with/without SARS-CoV-2 infection (hazard ratio adjusted for age: 1.41). Trilla et al. [44] studied 124 COVID-19 pregnant patients infected in the first trimester compared to 693 non-infected pregnant women. They indicated no significant difference in the risk of developing pre-eclampsia between the two groups (*p* = 0.422). Guida et al. [19] also observed the same result; pre-eclampsia occurred in 10.3% vs. 13.1% of pregnant women with and without COVID-19, respectively (*p* = 0.41). Mullins et al. [45] also reported the same result. In a cohort study on 64 pregnant patients with COVID-19, only two patients developed pre-eclampsia [46]. Hill et al. calculated a relative risk of 0.98 for pre-eclampsia and 0.88 for gestational hypertension among COVID-19 patients [21].

In a case series by Juusela et al., one of the presented COVID-19 pregnant patients developed pre-eclampsia [47]. In the study reported by Askary et al. [48], two maternal deaths occurred due to pre-eclampsia in a series of 16 COVID-19-positive pregnant women.

In a case reported by Ahmed et al., SARS-CoV-2 is a risk factor for pre-eclampsia and hypertension in a pregnant patient [49]. Azarkish et al. observed a case of a pregnant woman with severe COVID-19 developing pre-eclampsia [50]. The study published by Hansen et al. described a case of a pregnant woman diagnosed with COVID-19 and pre-eclampsia together at the same time [51]. Naeh et al. reported a case of a COVID-19-positive pregnant woman with secondary hypertension [52] (Table 1 and Table 3).

### 3.4. Cardiomyopathy, Myocardial Infarction, Heart Failure, and ECG Changes

Yousefzadeh et al. assessed the changes in the ECG of 89 pregnant women with coronavirus disease in their retrospective study [53]. Based on their report, 64 patients manifested normal ECG, AV block was observed in 3.4% of patients, and 17% of patients showed long QTC intervals. In a group of 154 pregnant coronavirus patients studied by Mercedes et al. [54], 15 (9.7%) developed myocardial injury. All 15 patients manifested left ventricular dysfunction with a mean ejection fraction of 37.67.

Mousavi Seresht et al. described four cases of cardiomyopathy in pregnant women associated with COVID-19. Two cases underwent heart transplant [55]. Juusela et al. presented two cases of COVID-19-related cardiomyopathy in pregnancy. These two severe COVID-19 patients developed cardiac dysfunction with moderately decreased left ventricular ejection fractions (40% and 45%) and global hypokinesis [47]. Moreover, Bhattacharyya et al. reported a case of Takotsubo cardiomyopathy at the 38th week of pregnancy [56]. In a case of a young pregnant woman reported by Nejadrahim et al., cardiomiopathy with a left ventricular ejection fraction of 40% was observed [57]. Soofi et al. reported acute heart failure and venous thrombosis in a case of a 25-year-old pregnant women infected with SARS-CoV-2. Significant dysfunction of the left ventricle during systole with an ejection fraction of only 15% was observed in this case [58]. Another peripartum cardiomyopathy case was reported by De Vita et al. The presented case also had clinical manifestations of congestive heart failure that were consistent with radiologic findings. A chest CT scan indicated interstitial and alveolar thickening in the right middle and inferior lobes, cardiomegaly, and bilateral pericardial and pleural effusion. Cardiovascular magnetic resonance imaging (CMR) demonstrated an oval-shaped, enlarged left ventricle with normal thickness, left ventricle walls diffuse hypokinesis with reduced systolic function, enlargement of the right ventricle with diffuse hypokinesis, and impaired contractile function. Echocardiography showed a severely dilated heart with reduced ejection fraction [16]. In the case reported by Franca and colleagues [59], a pregnant woman with a history of heart disease underwent premature labor due to COVID-19 infection, which resulted in the death of the mother. Gulersen et al. reported a case of a COVID-19-positive pregnant woman with myocarditis and an elevation of inflammatory markers. Khodamoradi et al. observed acute pulmonary embolism and COVID-19 in a postpartum patient [60]. In the case reported by Mevada et al., a pregnant patient presented COVID-19 and cardiomyopathy together. Cardiomyopathy and heart failure were observed in a pregnant woman with confirmed COVID-19 infection in the study conducted by Stout et al. [61] (Table 1 and Table 3).

### 3.5. Thrombotic Events in COVID-19 Patients during Pregnancy

The SARS-CoV-2-positive pregnant patient reported by De Vita et al. also had thrombosis in a branch of the pulmonary artery and the left ventricular apex [16]. Agarwal et al. reported a pregnant COVID-19 case who experienced portal hypertension due to portal vein thrombosis (PVT). Khodamoradi et al. observed acute pulmonary embolism and COVID-19 in a postpartum patient [62]. Zarrintan et al. reported a case of a 39-year-old pregnant woman manifesting pulmonary thromboembolism and severe COVID-19 [63]. Goudarzi et al. presented a case of pulmonary embolism in a 22-year-old pregnant woman with coronavirus disease [64] (Table 1 and Table 3).

### 3.6. Heart Rate Disorders

In the study conducted by Sinaci et al., tachycardia and bradycardia were observed in 5.4% and 1.3% of COVID-19 pregnant patients, respectively [65]. In a case series of seven patients, one pregnant women with coronavirus disease manifested tachycardia [66]. Tachycardia was observed in five out of 16 pregnant women with COVID-19-positive infection in the case series reported by Askary et al. [48]. Pachtman Shetty et al. recorded bradycardia and maternal heart rate nadir among the primary outcomes of pregnant patients with COVID-19 [17]. Donovan et al. presented a case of fetal heart rate acceleration in a COVID-19 pregnant mother [67] (Table 1 and Table 3).

### 3.7. Doppler Findings and Fetal Complications

A prospective study conducted by Rizzo et al. [68] on 54 infected and 108 non-infected pregnant women indicated no statistically important difference in umbilical vein blood flow/abdominal circumference (UVBF/AC) (*p* = 0.07) in the left and right Atrial Area (AA) (left 1.30 vs. 1.28 *p* = 0.221 and right 1.33 vs. 1.31 *p* = 0.324), as well as in the ventricular sphericity indices (SI) (left 1.75 vs. 1.77 *p* = 0.208 and right 1.51 vs. 1.54 *p* = 0.121) between the two groups. Fetal cardiac output (CO) was evaluated in 48 pregnant women with severe COVID-19 and 50 controls by Turgut et al. [69]. They demonstrated that fetal left CO dramatically decreased in women with severe disease after recovery compared to the controls (285.1 ± 114.7 vs. 246.7 ± 109.0 *p* = 0.058). Sule et al. reported fetal pulmonary artery Doppler parameters in 55 pregnant women with COVID-19 and 93 controls. Acceleration time (AT) was higher in the COVID-19-positive group than in the control group controls (54 ± 15.4 vs. 50 ± 14.9; *p* = 0.044). However, the ejection time (ET) (223 ± 45.4 vs. 220 ± 41.3; *p* = 0.960) and acceleration/ejection time ratio (PATET) (0.25 ± 0.07 vs. 0.23 ± 0.07; *p* = 0.191) parameters were not significantly different between the groups [70]. A study by Anuk et al. [71] indicated that the pulsatility and resistance indices (PI and RI) of both the umbilical and uterine artery had a significant increase in pregnant cases that were recovered from COVID-19 in comparison with the controls (*p* < 0.05). Moreover, in a multivariable logistic regression analysis, the PI and RI of the mean uterine artery were independently associated with disease (*p* = 0.009 and *p* = 0.049, respectively).

Mercedes et al. [54] reported that among 15 pregnant COVID-19 infected pregnant women with myocardial injury, three (20%) developed fetal bradycardia. In a cohort study by Adhikari et al. [65], 27% of patients and 33% of controls underwent caesarean delivery, for which the reason was abnormal fetal heart rate in 3% of patients and 5% of controls. The prevalence of abnormal fetal heart rate was not significantly different between patients and controls (*p* = 0.13). In the study by Soto-Torres et al. [10], abnormal Doppler findings were present in eight of 106 patients (7%) vs. two of 103 controls (2%). There was no statistically significant difference between groups (*p* = 0.08). Doppler parameters were not significantly different between patients and controls. The umbilical artery PI was 0.67 (sd = 1.49) among patients and 0.4 and (sd = 1.04) among controls (*p* = 0.22). Gracia-Perez-Bonfils et al. studied 12 patients with coronavirus disease and evaluated the cardiotocograph of their fetuses. An increased baseline fetal heart rate (more than 10%) was observed in all fetuses [72].

Kato et al. reported tachycardia in the fetus of a pregnant woman with COVID-19 [73]. On the other hand, Cetera et al. reported bradycardia in the fetus of the three reported pregnant COVID-19 cases [74]. Pelayo et al. presented a case of a pregnant COVID-19 patient who underwent caesarean due to a non-reassuring heart rate of the fetus [75]. Gubbary et al. presented a case of fetal myocarditis associated with COVID-19 with an ejection fraction of 35%, which was followed by non-immune hydrops [76]. (Table 1, Table 2, Table 3 and Table 4).

**Table 1 jcm-11-06194-t001:** Characteristics and maternal outcomes of the included cohort/cross-sectional/case-control studies.

First Author	Year	Study Design	Country	Number of Cases	Number of Controls	Pregnancy Phase	Mean Age of Cases	Mean Age of Controls	Cardiovascular Comorbidities Among Cases (Percentage)		Maternal Cardiovascular Complications Among Cases (Percentage)				
									Hypertension	Others	Pre-eclampsia/Eclampsia/Gestational Hypertensive Disorders	Arrhythmia (Type)	Thrombotic Events	Doppler Findings	Others
Kalahroudi et al. [28]	2021	Cohort	Iran	56	94	31.9 ± 8.22 weeks	31.6 ± 6.1 years		6 (10.7%)		10 (19.8%)				Pre-term labor (34.5%), caesarean delivery (67.3%)
Antoun et al. [30]	2020	Cohort	UK	23	6756	Third trimester *n* = 19 Second trimester *n* = 4	29.0				2/23 (8.7%)				Pre-term labor 7/19 (36.4%), caesarean delivery 16/19 (68.4%)
Arslan et al. [38]	2021	Cohort	Turkey	110	6508	Third trimester *n* = 106 (96.4%) Second trimester *n* = 4 (3.6%)	31.0 ± 6.0 years				11 (10%)				Pre-term labor *n* = 37 (33.6%)
Melguizo et al. [20]	2021	Cohort	Spain	1347	1607		33.0	33.0	19/1304 (1.5%)	Baseline heart disease 15/1316 (1.1%)	69 (5.1%) (Severe pre-eclampsia *n* = 28/69 (40.6%))		Venous thrombotic events (1.5%), DVT *n* = 10 (0.7%), pulmonary embolism *n* = 4 (0.3%)	Positive ultrasound prematurity screening *n* = 16 (1.4%)	Pre-term delivery (11.1%), premature rupture of membranes (15.5%), maternal mortality *n* = 2 (0.1%)
Daclin et al. [42]	2021	Case-control	France	86	86	39.29 ± 2.65 weeks	32.1	31.9			58/3235 (1.7%)				Pre-term delivery 6.2%, stillbirth *n* = 14/3235 (0.4%)
Epelboin et al. [31]	2021	Cohort	France	874	243,771	Third trimester	31.1 ± 5.9	30.5 ± 5.4	17/874 (1.9%)		42/874 (4.8%)				Pre-term birth *n* = 146/874 (16.7%), caesarean delivery *n* = 288/847 (32.9%), mortality *n* = 2/874 (0.2%), gestational hypertension *n* = 20/874 (2.3%)
Ferrera et al. [43]	2022	Cohort	USA	1332	42,554		28.8 ± 5.5	30.8 ± 5.2	301/1332 (22.6%)		20/1332 (1.5%)		Venous thromboembolism *n* = 4/1332 (0.3%)		Pre-term birth *n* = 143/1332 (10.7%), gestational hypertension *n* = 34/1332 (2.6%), caesarean delivery *n* = 357/1332 (26.8%), stillbirth *n* = 9/1332 (0.7%)
Guida et al. [19]	2022	Case-control	Brazil	21 (COVID-19 with pre-eclampsia) 10.3%	182 (COVID-19 without pre-eclampsia) 89.7%	Third trimester 131 Second trimester 49 First trimester 23	>35: *n* = 7/21 =<35: *n* = 14/21		7/21 (33.4%)		Eclampsia *n* = 1/21 (0.5%), imminent eclampsia *n* = 5/21 (2.5%)				Maternal death *n* = 2/21 (9.5%), pre-term birth *n* = 10/21 (47.6%/), caesarean delivery *n* = 19/21 (90.5%)
Gurol et al. [32]	2021	Cohort	England	3527	338,553		=<19: *n* = 94/3527 20–24: *n* = 581/3527 25–29: *n* = 1040/3527 30–34: *n* = 1079/3527 35–39: *n* = 587/3527 +40: *n* = 146/3527				139/3527 (3.9%)				Pre-term birth *n* = 369/3047 (12.1%), fetal death *n* = 30/3527 (0.85%), emergency caesarean delivery *n* = 975/3527, (27.6%)
Hill et al. [21]	2021	Cohort	USA	218	413	Third trimester	29.7	30.1	3/218 (1.4%)		9/218 (4.2%)				Gestational hypertension *n* = 7/218 (3.3%), pre-term delivery *n* = 50/218 (22.93%)
Jayaram et al. [27]	2021	Cohort	USA	75	334	Third trimester	27.0	28.0	12/75 (16.0%)		18/75 (24.0%)				Gestational HTN *n* = 8/75 (10.6%)
Ahlberg et al. [77]	2020	Cohort	Sweden	155	604		32.0	32.1			12/155 (7.7%)				Pre-term birth *n* = 14/155 (9%)
Mahajan [33]	2020	Retrospective study (historic cohort)	India	879	63	38 and 34.5 weeks	27.0				Pre-eclampsia *n* = 50 (5.68%)/, eclampsia *n* = 4 (0.45%)				Dyspnea *n* = 38 (4.32%)
Melguizo [20]	2020	Cohort	Spain	1347	1607	39 weeks and 2 days	33.0	33.0	19 (1.5%)	Congenital heart disease *n* = 15 (1.1%)	Pre-eclampsia *n* = 69 (5.1%)		Thrombotic events: *n* = 7 (0.5%), deep vein thrombosis *n* = 10 (0.7%), pulmonary embolism *n* = 4 (0.3%)		
Mendoza [39]	2020	Cohort	Spain	8	34	Third trimester, 31.6 weeks	39.4	30.9			Pre-eclampsia *n* = 5 (62.5%)			UtAPI *n* = 1 (12.5% of cases and 2.3% of total)	
Mercedes et al. [54]	2020	Cohort	Dominican Republic	15	139	32.31 weeks	29.8	29.8				Palpitation *n* = 2 (13.3% of cases), atrial fibrillation and torsades de pointes and ventricular tachycardia *n* = 2 (13.3%), sinus tachycardia *n* = 4 (26.6%)			Elevated BNP and troponin in all cases group *n* = 15 (100%), BBB *n* = 5 (33.3%), ST depression and T-invert *n* = 6 (40%)
Molteni [78]	2020	Replication cohort	USA, UK and Sweden	16,718	1,328,248	14% in the first trimester, 43% in the second trimester, and 43% in the third trimester	31.0	31.0							Dyspnea *n* = 20/29 (73.6%), chest pain: 17 of 29 (62.3%)
Mullins et al. [45]	2020–2021	Prospective web-based registry—an observational cohort study	UK, Italy, China, Greece, Indonesia, India, Argentina, China, Czech Republic, Albania, Austria, Egypt and Chile	5824	1923		34.0	34.0			Pre-eclampsia *n* = 389/8189 (4.8%), eclampsia *n* = 41/8192 (0.5 %)				
Chavan et al. [79]	April 2020–June 2021	Prospective observational (cohort/cross sectional)	India	17	Not mentioned (but the whole numbers of data were 460)	Not mentioned	26	26			Pre-eclampsia *n* = 6/17 (35.2 %)				
Osaikhuwuomwan et al. [40]	2020	Cross-sectional	Nigeria	19	69	30.4 weeks	31.4	31.4			Pre-eclampsia/eclampsia *n* = 5/19 (26.3%)				
Palomo et al. [80]	2020	Cohort	Spain	17	10	39.1	35.6	36.9	Hypertension *n* = 2 (11.7% of cases)						
Papageorghiou et al. [34]	2021	Cohort	Argentina, Brazil, Egypt, France, Ghana, India, Indonesia, Italy, Japan, Mexico, Nigeria, North Macedonia, Pakistan, Russia, Spain, Switzerland, the United Kingdom, and the United States	725	1459		29.95	30.31	*n* = 26/725 (3.58%)		Pre-eclampsia *n* = 59/725 (8.13%)				
Pierce-Williams et al. [46]	2020	Cohort	USA	44	20	32 weeks	33	33	Chronic hypertension and cardiomyopathy *n* =11 (17%)		Pre-eclampsia *n* = 2 (3%)				Cardiac arrest *n* = 1 (3%)
Pirjani et al. [35]	2020	Cohort	Iran	66	133	32.64 weeks	30.97	28.79	-		-	Tachycardia *n* = 5 (7.57 %)		-	Dyspnea *n* = 27 (40.9%)
Serrano et al [36]	2021	Observational retrospective study (cross-sectional)/ historic cohort	Spain	231	13,033	First trimester (0–12 weeks)	31.8	32.6	Chronic hypertension *n* = 6/218 (2.8%), history of pre-eclampsia *n* = 9/218 (4.1 %)	High risk for pre-eclampsia *n* = 44/23 (19%)					
Trilla et al. [44]	2020	Prospective population-based study (cohort)	Spain	124	639	First, second and third trimester	33.1	33.9	Chronic hypertension *n* = 4/124 (3.2%)	pre-eclampsia *n* = 8/113 (7.3%)					
Yousefzadeh et al. [53]	2020	Cross-sectional	Iran	89	111	Mean = 29.59 weeks third trimester	31	31	-		-	Bradycardia *n* = 4 (4.5%), tachycardia *n* = 18 (20.5%)			ECG findings: atrioventricular (AV) block *n* = 3 (3.4%), first-degree AV block type (PR interval >200 ms) *n* = 3 (3.4%), mean QTC interval = 428.6 ± 37.4 ms, long QTC intervals (QTC ≥ 460 ms) *n* = 15 (17%), bundle branch block (QRS > 100 msec) (4.4%) of which three patients were RBBB and one was LBBB
Adhikari et al. [26]	2020	Cohort	USA	245	3035	First, second and third trimester	27	27			26 (11%)				
Anuk et al. [71]	2021	Case-control	Turkey	30	40	Median = 31.5	Median = 30	Median = 29							Increased PI and RI of uterine artery, mean uterine artery PI = 1 (sd = 1), mean uterine artery RI = 0.6 (sd = 0.2), pre-term delivery *n* = 6 (20%)
Wu et al. [23]	2020	Cohort	China	29	-		29			2 (6.90%)					
Pachtman et al. [17]	2021	Cohort	USA	20	-					5 (25%)		Bradycardia (defined as <60 bpm), and maternal heart rate (HR) nadir *n* = 10/31 (32%)			Elevated cardiac troponins (I, T, or high sensitivity) *n* = 4/18 (22%), elevated BNP *n* = 3/10 (30%)
Berengueet et al. [18]	2020	Cohort	Spain	4035					2052/4010 (51.2%)	Chronic heart disease *n* = 932/3994 (23.3%)					
Brandt et al. [81]	2021	Case-control	USA	61	122				2 (3.3%)						
Jering et al. [37]	2021	Cohort	USA	6380	400066				288 (4.5%)		Pre-eclampsia *n* = 564 (8.8%), eclampsia *n* = 8 (0.1%)		VTE *n* = 15 (0.2%), thrombotic event *n* = 22 (0.3%)		Myocardial infarction *n* = 8 (0.1%)
Soto-Torres et al. [10]	2021	Case-control	USA	106	103						Pre-eclampsia *n* = 21 (19.8%), other hypertensive disorders *n* = 17 (16%)				
Rosenbloom et al. [41]	2021	Cohort	Israel	83	166	39 w	26	28	10 (12.1%)		Gestational hypertension *n* = 10 (12.1%), pre-eclampsia *n* = 14(16.9%), other hypertensive disorders *n* = 24 (28.9)				

DVT = deep vein thrombosis, BBB = bundle branch block, PI = pulsatility indices, RI = resistance indices, UtA = uterine artery, VTE = venous thromboembolism, sd = standard deviation.

**Table 2 jcm-11-06194-t002:** Characteristics and fetal outcomes of the included cohort/cross-sectional/case-control studies.

First Author	Year	Study Design	Country	Number of Cases	Number of Controls	Pregnancy Phase	Mean Age of Cases	Mean Age of Controls	Cardiovascular Comorbidities among Cases (%)		Fetal Cardiovascular Complications among Cases (%)			
									Hypertension	Others	Arrhythmia (Type)	Thrombotic Events	Cardiomyopathy/Heart Failure	Doppler Findings
Ayhan et al. [82]	2022	Case-control	Turkey	45	45	32 ± 4 weeks		33 ± 1 weeks						Aortic peak velocity = 79.4 cm/s, pulmonary peak velocity = 67.4 cm/s, left myocardial performance index = 0.57 ± 0.14, tricuspid annular plane systolic excursion (TAPSE) = 6.87 mm, mitral annular plane systolic excursion (MAPASE) = 6.69 mm
Mercedes et al. [54]	2020	Cohort	Dominican Republic	15	139	32.31 week	29.87	29.87		-	Fetal bradycardia *n* =3 (20% of patients)			
Sinaci et al. [65]	2020	Cohort	Turkey	224	Over 150,000	37 weeks	28	Not mentioned		-	Tachycardia *n* = 12 (5.4%), bradycardia *n* = 3 (1.3%)			
Adhikari et al. [26]	2020	Cohort	USA	245	3035		27	27	26 (11%)		Abnormal fetal heart rate *n* = 7 (3%)			
Anuk et al. [71]	2021	Case-control	Turkey	30	40	Median = 31.5	30	29						Increased PI and RI of umbilical arteries, mean umbilical artery PI = 1 (IQR = 0.2), mean umbilical artery RI = 0.6 (IQR = 0.1), mean uterine artery PI = 1 (IQR = 1), mean uterine artery PI = 0.6 (IQR = 0.2)
Soto-Torres et al. [10]	2021	Case-control	USA	106	103						Fetal tachycardia *n* = 1 (1%), premature atrial contractions *n* = 1 (1%)			Abnormal Doppler findings *n* = 8 (7%), mean PI of umbilical artery = 0.67 (No significant difference between patients and controls)
Sule et al. [70]	2021	Cohort	USA	55	93									Higher AT of pulmonary artery in patients, no significant difference in ET and PATET of pulmonary artery between patients and controls

PI = pulsatility indices, RI = resistance indices, AT = acceleration time, ET = ejection time, PATET = acceleration/ejection time ratio.

**Table 3 jcm-11-06194-t003:** Characteristics and maternal outcomes of the included case report/case series studies.

First Author	Year	Study Design	Country	Number of Cases	Pregnancy Phase	Mean Age	Cardiovascular Comorbidities		Maternal Cardiovascular Complications					
							Hypertension	Others	Pre-eclampsia/Eclampsia	Arrhythmia (Type)	Thrombotic Events	Cardiomyopathy/Heart Failure	Doppler Findings	Others
Ahmed et al. [49]	2020	Case report	UK	1	37 weeks	26	173/111 mmHg		Pre-eclampsia					HELLP syndrome
Askary et al. [48]	2020	Case series	Iran	16	Third trimester 10, second trimester 5, first trimester 1	30.06			7/16		Pulmonary thromboembolism *n* = 1	Massive myocardial infarction *n* = 1		Pre-term labor *n* = 6/16, caesarean delivery *n* = 9/12, premature labor pain (PLP) *n* = 2, placenta accreta *n* = 2
Azarkish et al. [50]	2021	Case report	Iran	1	38 weeks	19	140/90 mmHg			Sinus tachycardia	Elevated PTT	Cardiorespiratory arrest and death		
Breslin et al. [66]	2020	Case series	USA	7	33.2 weeks	33.85						Bronchospasm, tachycardia		Hypertension
Seresht et al. [55]	2022	Case series	Iran	4	25 weeks	29.25		Large atrial septal defect		Tachycardia 4/4		(1) Minimal pericardial effusion, EF 25%, heart failure (2) Severe acute cardiac failure, acute left ventricular failure, EF 20% (3) EF 30%, pulmonary edema (4) Heart failure, EF 35%	Severe mitral valve regurgitation	Intrauterine fetal demise with oligohydramnios, expired *n* = 2/4
Donovan et al. [67]	2022	Case report	USA	1	35 weeks and 5 days	36				Tachycardia 110 beats/min				
Franca et al. [59]	2022	Case report	Brazil	1	28 weeks	26		Mitral valve replacement in 2011 due to mitral valve stenosis		Tachycardia >145 beats/minute, ventricular tachyarrhythmia, mitral murmur				High blood pressure (147/95 mmHg), death
Goudarzi et al. [64]	2022	Case report	Iran	1	Third trimester	22					Massive pulmonary embolism	Right and left-sided heart failure		First echocardiography showed a very dilated right atria and ventricle fetal death. Second echocardiography showed right ventricle enlargement, severe dysfunction, McConnell’s sign, moderate tricuspid regurgitation, dilated pulmonary artery, mild pulmonary insufficiency, dilated inferior vena cava, respiratory cardiovascular arrest and death
Gulersen et al. [60]	2021	Case report	USA	1	28 weeks and 4 days	31			Yes	Tachycardia (heart rate 122 beats per minute), sinus tachycardia	Multiple risk factors for venous thromboembolism	Pericardial effusion biventricular dysfunction		Caesarean birth
Hansen et al. [51]	2020	Case report	USA	1	34 weeks	31	162/86 mm Hg		Presumed pre-eclampsia					Caesarean birth
Khodamoradi et al. [62]	2020	Case report	Iran	1	37 weeks and 2 days	36					Pulmonary embolism, hypercoagulable state			Had elective scheduled caesarean section
Leal et al. [25]	2020	Case series	Brazil	5469	First trimester: *n* = 378 6.9 %, second trimester: *n* = 985 18%, third trimester: *n* = 2475 45.25%, postpartum and postabortion: *n* = 1393 25.4%	30	*n* = 562 (10.2%)							Chest pain: *n* = 107 (2%)
Naeh et al. [52]	2021	Case report	Canada	1	28 week (third trimester)	39	152/132 mm Hg			Sinus tachycardia PR = 141				Dyspnea
Nejadrahim et al. [57]	2020	Case report	Iran	1	Postpartum	38			Severe pre-eclampsia	Tachycardia PR = 115		Postpartum cardiomyopathy/heart failure with LVEF of 30%, global hypokinesis, LV enlargement (LV end diastolic size 5.8 cm)		Dyspnea/ pulmonary edema
Pelayo et al.	2020	Case report	USA	1	36 week and 2 days	35				Tachycardia		Cardiomyopathy as evidenced with 45% ejection fraction on echocardiography		Pulmonary embolism
Radoi et al. [83]	2021	Case report	Romania	1	26 weeks	19			Pre-eclampsia BP = 178,110	Tachycardia		Heart failure with ejection fraction less than 30%		Dyspnea, cyanosis, pulmonary edema, chest pain, pulmonary hypertension, VSD and Mirror and Eisenmenger Syndrome, increased jugular venous pressure
Soofi et al. [58]	2020	Case report	Saudi Arabia	1	34 weeks	25				Sinus tachycardia and occasional PVCs	DVT in the right superficial femoral Vein	Acute heart failure on echocardiography exhibited features of severe left ventricle systolic dysfunction. Cardiac MRI displayed severe global LV and RV systolic dysfunction,		Pleuritic chest pain with dyspnea
Stout et al. [61]	2020	Case report	UK	1	36 weeks and 4 days	28				Tachycardia PR = 120, ventricular fibrillation, ventricular escape rhythm	-	-	-	Cardiac arrest
Vaezi et al. [24]	2020	Case series	Iran	24	Third trimester (15 cases, 62.5%), 6 cases were in their second trimester (25%), and only 3 cases (12.5%) were under 14 weeks	26.5	*n* = 1 (4.16%)	HELLP *n* = 1 (4.16%)	Severe pre-eclampsia *n* = 3 (12.5%)	Tachycardia *n* = 3 (12.5%)				Dyspnea *n* = 10 (41.6 %)
Zarrintan et al. [63]	2020	Case report	Iran	1	34 weeks	39				Sinus tachycardia	Pulmonary thromboembolism	Echocardiography showed an ejection fraction of 45–50%. Severe right ventricle enlargement was also evident, with moderate to severe dysfunction of the right ventricle.		Dyspnea, chest discomfort. ECG showed an S wave in the lead I, q wave in lead II and sinus tachycardia, suggestive of right ventricle straining.
Agarwal et al. [84]	2021	Case report	India	1	Third trimester	28								Portal hypertension due to portal vein thrombosis (PVT)
Bhattacharyya et al. [56]	2020	Case report	India	1	38 w							Takotsubo cardiomyopathy		
De Vita et al. [16]	2020	Case report	Italy	1	Third trimester	35					Thrombosis both in the apical left ventricle and in a branch of the pulmonary artery	Peripartum cardiomyopathy, congestive heart failure		
Juusela et al. [47]	2020	Case series	USA	2	39 w 2 d–33 w 6 d				Pre-eclampsia			Cardiomyopathy (cardiac dysfunction with moderately reduced left ventricular ejection fractions of 40% and 45% and hypokinesis)		

DVT = deep vein thrombosis, LV = left ventricle, RV = right ventricle, VSD = ventricular septal defect.

**Table 4 jcm-11-06194-t004:** Characteristics and fetal outcomes of the included case report/case series studies.

First Author	Year	Study Design	Country	Number of Cases	Pregnancy Phase	Mean Age	Fetal Cardiovascular Complications			
							Arrhythmia (Type)	Thrombotic Events	Cardiomyopathy/Heart Failure	Others
Cetera et al. [74]	2021	Case series	Italy	3			Fetal bradycardia 1			Type 2 fetal cardiotocography *n* = 1 fetal demise *n* = 2
Perez et al. [72]	2020	Case series	Spain	12	37.80 weeks		Gross fetal tachycardia (>210 bpm) *n* = 1/12, absence of accelerations *n* = 12/12, prolonged decelerations *n* = 10/12, zigzag pattern *n* = 4/12 (33%)			Absence of cycling *n* = 7/12 (58.3%)
Gubbari et al. [76]	2022	Case report	India	1	34 weeks	17 days	Persistent tachycardia		Right ventricular dilation	Echocardiography revealed pulmonary hypertension biventricular hypertrophy, severe mitral and tricuspid regurgitation with EF = 35% on day 2
Kato et al. [73]	2022	Case report	Japan	1	21 weeks				Tachycardia (165 bpm)	Intrauterine fetal death The RT-PCR test result from a dead neonatal nasopharyngeal swab was positive
Mevada et al. [85]	2020	Case report	India	1	Third trimester (GA: 36.5 by date and 37 by scan)	30	Tachycardia (PR = 110)		PPCM or viral cardiomyopathy	Echocardiography: global left ventricular hypokinesia, EF 25–30%
Radoi et al. [83]	2021	Case report	Romania	1	26 weeks	19				Hydrops fetalis, Mirror syndrome
Vaezi [24]	2020	Case series	Iran	24	Third trimester (15 cases, 62.5%), 6 cases were in their second trimester (25%), and only 3 cases (12.5%) were under 14 weeks	26.5	Fetal bradycardia *n* = 1 (4.16%)			
Mongula et al. [86]	2020	Case report	Netherlands	1	31w4d		Fetal tachycardia			
Wang et al. [87]	2022	Case report	USA	1	31w5d		Supraventricular tachycardia			

PPCM = post-partum cardiomyopathy, EF = ejection fraction.

## 4. Discussion

This review article showed that COVID-19 infection could potentially be associated with some cardiovascular outcomes among pregnant women. According to the reviewed articles, pre-eclampsia, eclampsia, hypertensive disorders, cardiomyopathy heart failure, MI thrombosis, and bradycardia were reported as cardiovascular complications among the COVID-19 infected pregnant women [26,47,56]. Moreover, there was evidence of increased pulsatility and resistance indices of the umbilical and uterine artery in maternal-fetal Doppler patterns among pregnant women who recovered from COVID-19 [71]. In addition, there was evidence of cardiovascular complications in the fetus of infected pregnant women, including arrhythmia and a higher AT of the pulmonary artery in the Doppler study [86].

During pregnancy, the cardiovascular system undergoes several changes. Cardiac output increases by 20% until eight weeks of gestational age. Peripheral vasodilation is mediated by factors such as nitric oxide, which is upregulated by vasodilatory prostaglandins (PGI2) and estradiol. Due to this action, systemic vascular resistance falls by 25–30%, and for compensation, during pregnancy, cardiac output increases by about 40%. This action happens by an increase in stroke volume, but heart rate also increases somewhat during pregnancy. Cardiac output reaches its maximum at about 20–28 weeks gestation and falls somewhat at term [88].

Researchers observed that SARS-CoV-2 could result in a spectrum of cardiovascular diseases, including thromboembolism, arrhythmia, myocarditis, and acute coronary syndrome [89,90,91]. The cardiac injury caused by COVID-19 results from cytokine storm, viral myocarditis, and ischemia [89,90,92]. Cardiac magnetic resonance (CMR) studies illustrated the indications of myocardial inflammation, fibrosis, left ventricle (LV) enlargement, and biventricular involvement in COVID-19 [93,94,95]. Similarly, normal pregnancy has multiple cardiovascular effects, including increased heart rate, cardiac output, and vascular volume; dyspnea on exertion; ejection murmurs over the pulmonary artery and aorta; and decreased venous return to the heart, which may lead to a presyncope condition [89,90,91,96,97,98].

The previous reviews reported 41 to 47% of pre-term labor among infected pregnant women [99,100]. According to a meta-analysis by Di Mascio et al., among hospitalized mothers infected with coronavirus, the most prevalent adverse pregnancy outcome was pre-term birth. Their results show that COVID-19 infection during pregnancy was significantly associated with higher rates of pre-term birth, pre-eclampsia, cesarean, and perinatal death [101]. Moreover, Lassi et al. reported a 2.4 fold higher risk of pre-term birth among pregnant cases in comparison with controls [102].

The SARS-CoV-2 virus enters the host cells via the receptor-angiotensin-converting enzyme2 (ACE-2), leading to dysfunction of the renin-angiotensin-aldosterone system (RAAS) [103]. The ACE-2 is an upregulated receptor during normal pregnancy [98]. This upregulation results in the modification of angiotensin II, which is a vasoconstrictor agent, to angiotensin-(1–7); these are vasodilators and can lower the blood pressure during normal pregnancy [104]. However, SARS-CoV-2, which binds to ACE-2, downregulates the level of ACE-2 and angiotensin-(1–7), which eventuates vasoconstriction, pre-eclampsia, uterine constrictions, and pre-term labor [104]. Moreover, endothelial cells express ACE-2 receptors [105], and endothelial cell infection and immune cell-mediated endothelial damage are reported in COVID-19 [106]. As endothelial failure is a key characteristic of pre-eclampsia, infection with SARS-CoV-2 during pregnancy may resemble or induce microvascular dysfunction by inducing endothelins [107]. Furthermore, the virus invades the cells, which over-activates the inflammatory response and increases TNF-α, IL-6, IL-1β, and monocyte chemoattractant protein-1 levels and establishes cytokine storm [65,108,109,110,111]. This hyperproduction increases the risk of vascular hyperpermeability and possibly causes hypertensive disorders [112]. These mechanisms can explain the mentioned hypertensive disorders among pregnant COVID-19 cases (Figure 2).

Another cardiovascular complication among pregnant COVID-19 cases was cardiomyopathy and heart failure. Heart failure as a potential complication of COVID-19 stems from various myocardial aggression mechanisms, including direct myocardial damage through viral action, direct and indirect inflammatory damage, imbalance of O2 supply, and inducing atherothrombotic events because of the inflammatory destabilization of atheromatous plaques, thus, causing acute myocardial dysfunction [113,114]. SARS-CoV-2 interacts with the myocardial tissue through binding the viral glycoprotein Spike 1 to ACE-2 receptors. ACE-2 receptors are especially expressed within cardiac pericytes. This interaction leads to a direct tissue injury [114]. Due to the higher presence of ACE2 receptors in postmortem cardiac pericytes isolated from patients with heart disease compared to those without prior disease, the concept of the cumulative effect of previous CV disease and troponin increase was proposed [115].

By multiple mechanisms, COVID-19 can lead to cardiac injury. Due to these mechanisms, extreme inflammatory response with myocarditis and endothelial injury occurs. In some situations during pregnancy, acute heart failure should be considered, including viral myocarditis, noncardiogenic pulmonary edema, and peripartum cardiomyopathy [116]. Peripartum cardiomyopathy (PPCM) is a potentially dangerous situation presenting in the last month of pregnancy as heart failure with reduced ejection fraction (HFrEF). The etiology of PPCM is not well known. Host susceptibility and systemic angiogenic imbalance are considered to be important in the pathophysiology of PPCM. Probable factors causing PPCM are viral infections, genetic predisposition, inflammation, low selenium levels, stress-activated cytokines, the induction of antiangiogenic factors, autoimmune reaction, pathological response to hemodynamic stress, and unbalanced oxidative stress [117].

Other cardiovascular events among pregnant COVID-19 cases were MI and thrombotic events. A German cohort study among non-pregnant patients reported that 71% of 100 patients had elevated cardiac biomarkers, including cardiac troponin (CTn) and pro-BNP, which indicate myocardial inflammation [118,119]. However, in a series of 31 positive SARS-CoV-2 pregnant women, 22% had elevated troponin levels, and 30% in BNP levels. Studies suggest that increased troponin and BNP levels are associated with malignant arrhythmia, ventricular tachycardia [120], and fibrillation [114,121].

It has been reported that thromboembolic complications are higher among infected pregnant women [122,123]. Acute viremia such as SARS-CoV-2 activates monocytes and macrophages, which trigger the blood coagulation system via producing interleukin 6 (IL_6) and tumor necrosis factor-α (TNF-α) [124]. Moreover, a recent study illustrates a connection between IL_6 and fibrinogen levels in patients with COVID-19 infection [125]. Furthermore, the activation of monocytes causes a loss of heparin sulfate from the endothelial surface, platelet activation, the downregulation of thrombomodulin, and nitric oxide production [126]. These contingencies lead to a prothrombotic state and increase the risk of VTE in infected patients. Thus, SARS-CoV-2 may directly attack the vascular endothelium, resulting in inflammation and the infiltration of inflammatory cells [127]. Eventually, this inflammation causes intravascular coagulation and thrombosis. A study showed increased levels of D-dimer among infected pregnant women. D-dimers are produced through fibrinolysis [128]. Subsequently, the D-dimer level was measured in infected pregnant women in the third trimester as a risk factor to evaluate the risk of thromboembolism compared to non-infected pregnant women [95]. However, the increased D-dimer level was strongly decreased after 2–3 days of anticoagulant therapy [95]. Meanwhile, in the coagulant therapy the risk of hemorrhage must be considered in pregnant women [129].

Discoagulation and vein thrombosis are also seen as cardiovascular damages in COVID-19 infection. The endothelial dysfunction caused by COVID-19 is developed by mediating leukocyte inflammation and adjusting the vessels’ integrity, leading to pro-coagulative condition, which is the leading cause of multi-organ damage such as cardiomyopathy during COVID-19 infection [130,131,132]. During normal pregnancy, a physiological hypercoagulation state is prepared for the bleeding caused by delivery [88]. There is an increased level of clotting factors, including factor VIII, IX, and X, as well as fibrinogen level [133].

On the other hand, the levels of anticoagulants such as antithrombin and protein S and the fibrinolytic activity are decreased. Since SARS-CoV-2 increases the production of inflammatory cytokines and activates the thrombin and coagulation system and clot formation, this can eventually lead to thrombosis [112]. The studies of Abouzaripour M. et al. and Martinelli et al. reported ovarian vein thrombosis and pulmonary emboli among COVID-19-positive pregnant women [134,135]. Thus, VTE risk assessment should be accomplished on all COVID-19 infected pregnant women, and VTE prophylaxis with LMWH is recommended for pregnant patients with moderate to severe COVID-19 infection [136,137]. Based on the guidelines for thromboprophylaxis in COVID-19 infected pregnant women, if the bleeding risk is low and labor is not expected within 12 h, LMWH is recommended for all hospitalized patients and patients with an increased VTE risk score (≥3) and should be continued for 10–42 days [138]. Moreover, a low-dose aspirin for the prevention of pre-eclampsia is prescribed for high-risk pregnant women. However, the risk of bleeding and emergency caesarean must be considered [139].

Furthermore, increased pulsatility and resistance indices of the umbilical and uterine artery in maternal-fetal Doppler patterns among pregnant women who recovered from COVID-19 were reported. A case report represented supraventricular tachycardia (SVT) in a fetus borne by a 26-year-old COVID-19-positive mother [87]. Fetus SVT is one of the most common arrhythmias in the presence of maternal COVID-19 infection [140]. Doppler ultrasound is used to assess fetal vessels’ well-being and is useful for predicting pre-eclampsia [141]. Khalil et al. found that the incidence of stillbirth was remarkably higher during the COVID-19 pandemic using uterine artery Doppler [142]. Furthermore, Ali T. Anuk and colleagues used Doppler ultrasonographic assessments of the uterine arteries (UTA) and umbilical artery among 30 COVID-19-positive pregnant women in their third trimester, which indicated a considerable increase in pulsatility and resistance of UTA and umbilical artery compared to the control group [71]. However, Ayhan et al. found no significant differences in the fetal Doppler parameters [143].

There are contradictory reports about vertical transmission. Fenizia et al. demonstrated that the SARS-CoV-2 genome had been discovered in umbilical cord blood, amniotic fluid, and maternal vaginal mucosa, although the infection of infants is rare. Further, if the newborns became infected, it was asymptomatic or mild in most cases [144]. However, the study of Corasso and colleagues determined both IgG and IgM antibodies against COVID-19 in seronegative newborns from infected mothers [145]. The source of these immunoglobulins might be the maternal blood, cord blood, and amniotic cord [127,146]. Moreover, research in China has not shown any virus in breast milk, but more studies are needed [147]. On the other hand, some studies did not discover any vertical transmission results [148]. Furthermore, in analyzing the coagulation indexes of these newborns, some had higher MYO, CK-MB, and D-dimer levels, which may lead to coagulopathy in neonates [149].

According to Shu Qin Wei’s study, severe COVID-19, compared to mild COVID-19, is strongly associated with pre-term birth, ICU admission, gestational diabetes, pre-eclampsia, caesarean delivery, NICU admission, mechanical ventilation, and low birth weight. The comparison of symptomatic and asymptomatic COVID-19 during pregnancy is that symptomatic COVID-19 increases the risk of caesarean delivery and pre-term birth, but it is not associated with gestational diabetes. They have also shown that SARS-CoV-2 infection during pregnancy, compared with healthy pregnant women, is associated with pre-term birth, pre-eclampsia, NICU admission, lower birth weight, ICU admission, and stillbirth. Compared with no infection, COVID-19 is not associated with postpartum hemorrhage, gestational diabetes, neonatal death, or caesarean delivery.

Cardiovascular complications also occur in the fetus of infected pregnant women, including arrhythmia and a higher acceleration time (AT) of the pulmonary artery in the Doppler study. The fetal pulmonary artery blood flow’s AT is an advent for monitoring pulmonary artery pressure [150,151]. Turgu et al. used pulmonary artery Doppler to assess the fetal lung development in women who had recovered from COVID-19 [152]. This study indicated that the main pulmonary artery peak systolic velocity was higher, and pulsatility, AT, and ejection time (ET) were remarkably higher in these women [152]. These Doppler evaluations may show insufficient lung development in the fetus and cause respiratory distress syndrome (RDS) in newborns [153].

There is a particular concern about the effectiveness and safety of COVID-19 vaccines in pregnant women. According to Prasad’s study, the effectiveness of mRNA vaccines in vaccinated pregnant women one week after a second dose against confirmed SARS-CoV-2 was 89.5%, and the chance of still birth was lower in this group. They also found no evidence of adverse outcomes such as placental abruption, miscarriage, pulmonary embolism, maternal death, neonatal intensive care unit admission, and postpartum hemorrhage in pregnant women that received mRNA COVID-19 vaccines [154].

Kirschbaum et al. reviewed seven women with heart disease (including congenital heart disease, acute myocarditis, and rheumatic valve disease) who were infected by COVID-19 during their pregnancies. Serious adverse outcomes were due to recurrent atrial flutter that caused hemodynamic instability, respiratory failure (ARDS), cardiogenic shock, and acute pulmonary edema. They also recommend that the overlapping of three conditions including pregnancy, heart disease, and COVID-19 is dangerous and pregnancy for women with heart disease should be planned after vaccination [155].

New drugs are widely used for the treatment of COVID-19, but our knowledge is limited to the usage of these drugs during pregnancy. Azithromycin is an antibiotic with anti-inflammatory, immunomodulatory, and potential antiviral effects. Although the role of this drug in COVID-19 treatment is not well established, the immunomodulatory effect of azithromycin can slow down cytokine storm. Prolonged QT interval is one of the side effects of azithromycin, which should be handled for pregnant women with heart disease during pregnancy [156].

The main limitation of this systematic review was the lack of studies investigating cardiovascular complications among pregnant COVID-19 cases. Therefore, we included case series and case reports as well. Another limitation was that several included studies did not provide sufficient data on the reported complications in details. For instance, regarding arrhythmia, we could not result which types of arrhythmia were more common among cases.

## 5. Conclusions

In conclusion, SARS-CoV-2 infection during pregnancy can potentially be associated with cardiovascular complications in mothers and their fetuses. Pre-eclampsia, eclampsia, hypertensive disorders, cardiomyopathy and heart failure, MI and thrombosis, and bradycardia were reported as cardiovascular complications among the COVID-19-infected pregnant women. Additionally, the increased pulsatility and resistance indices of the umbilical and uterine artery in maternal–fetal Doppler patterns among pregnant women who recovered from COVID-19 were reported. Moreover, there was some evidence of cardiovascular complications in the fetuses of infected pregnant women, including arrhythmia and a higher AT of the pulmonary artery in the Doppler study. Further research is needed to attain comprehensive information on the prevalence and risk of maternal and fetal cardiovascular complications during the COVID-19 infection pandemic.

## Figures and Tables

**Figure 1 jcm-11-06194-f001:**
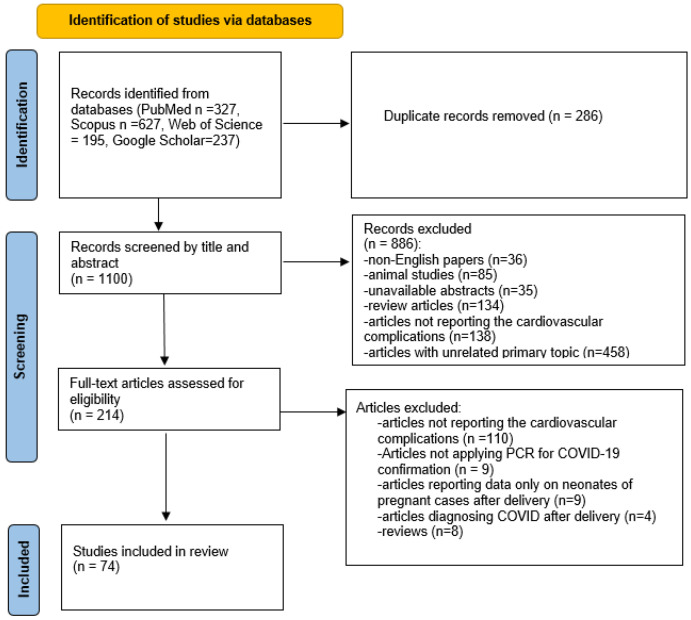
PRISMA flow diagram.

**Figure 2 jcm-11-06194-f002:**
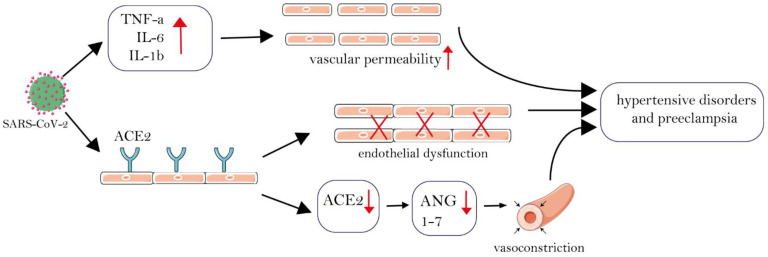
Association of SARS-CoV-2 with hypertensive disorders and pre-eclampsia. SARS-CoV-2 binds to ACE-2 and reduces ACE-2 and ANG-(1–7) levels. Reduced ANG-(1–7) decreases its vasodilatory effect and, therefore, enhances vasoconstriction. Endothelial cells also express ACE2 receptors and SARS-CoV-2 infection can cause immune cell-mediated endothelial damage, which is a hallmark of pre-eclampsia. Additionally, the virus invades the cells, activating the inflammatory response by increasing TNF-, IL-6, IL-1. This cytokine hyperproduction increases the risk of vascular hyperpermeability, which can lead to hypertensive diseases. These mechanisms could explain the hypertension problems seen in COVID-19 individuals who are pregnant. (ACE2= Angiotensin-converting enzyme 2, ANG = angiotensin, IL = interleukin, TNF-a = tumor necrosis factor α).

## Data Availability

This review study was not registered, and a protocol was not prepared. This study followed the PRISMA checklist. The template data-collection forms, data extracted from included studies, and any other materials used in this review were not publicly available.

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
