# Peer review of "Cardiovascular Complications of COVID-19 among Pregnant Women and Their Fetuses: A Systematic Review"

_jcm, 2022, doi:10.3390/jcm11206194_

Round 1
Reviewer 1 Report (Previous Reviewer 2)
The manuscript has improved significantly. I have no further comment.
Author Response
Many thanks for your opinion and attention.
Reviewer 2 Report (New Reviewer)
I read with interest the paper JCM 1949850 by Yagoobpoor S et al.
This is a systematic review on the cardiovascular complications of SARS-CoV2 infection in a particular subset of patients, namely pregnant women and their fetus.
My comments:
The methodology used to perform the systematic review is good. The document is well presented and easy to read, and the various parts each take up the space they deserve. The figures are well done.
Line 248: relative and not relative
Lines 265-270: the same concept is repeated in two different sentences.
Lines 298-313: "Thrombosis and coagulopathies" section
Ok for thrombosis, but why coagulopathy / thrombocytopenia / DIC in a review on "cardiovascular complications" and not on "hematological complications". I would delete the part relating to anything that is not strictly cardiovascular (therefore please review the studies excluded or included in the systematic review).
The 4 tables are illegible as such. Please format it so that it is readable in limited space. Shouldn't they be paginated where they are mentioned?
Author Response
Reviewer 2:
I read with interest the paper JCM 1949850 by Yaghoobpoor S et al.
This is a systematic review on the cardiovascular complications of SARS-CoV2 infection in a particular subset of patients, namely pregnant women and their fetus.
The methodology used to perform the systematic review is good. The document is well presented and easy to read, and the various parts each take up the space they deserve. The figures are well done.
Many thanks for your consideration and valuable opinion.
Line 248: relative and not relative
Thanks for your attention. We edited the typo and changed “ralative risk” to “relative risk”.
Lines 265-270: the same concept is repeated in two different sentences.
Thanks for your attention. We deleted the second sentence accordingly.
Lines 298-313: "Thrombosis and coagulopathies" section
Ok for thrombosis, but why coagulopathy / thrombocytopenia / DIC in a review on "cardiovascular complications" and not on "hematological complications". I would delete the part relating to anything that is not strictly cardiovascular (therefore please review the studies excluded or included in the systematic review).
Thanks for your comment. We deleted the parts related to coagulopathy, thrombocytopenia, and DIC in the text and tables. Therefore, four studies were excluded and we changed the PRISMA diagram and study selection section accordingly.
The 4 tables are illegible as such. Please format it so that it is readable in limited space. Shouldn't they be paginated where they are mentioned?
Many thanks for your valuable comment. We edited and re-formatted the tables and attempted to make them readable in a more limited space. Also, we refer to the tables in the text where they were mentioned accordingly.
This manuscript is a resubmission of an earlier submission. The following is a list of the peer review reports and author responses from that submission.
Round 1
Reviewer 1 Report
I don't think the manuscript offers anything new in its current form. Indeed, the lack of extensive clinical trials and clear recommendations make the subject difficult to approach. This article focuses on an important topic related to the clinical implications.
However, some suggestions could improve the quality of the article:
- Line 29 myocardial infarction (MI) – why was the abbreviation made?
- Restructure the introduction
- Line 131 The mean age of cases 131 was in the range of 26 to 45 – please specify the mean value
- Line 133-135 Please specify only cardiovascular comorbidities
- In table 1, maternal and fetal complications must be differentiated; to make possibly two tables
- Section 3.3 The fetal outcomes? It is an enumeration of some studies without a clear structuring of some ideas
- Section 3.4 Please restructure this section
- Section 3.7 The change of IR and IP at the level of the umbilical artery, what repercussions do they have at the level of the fetus? Correlation with maternal PI and RI at uterine level
- Nothing related to the physiopathological mechanisms of maternal and fetal cardiac decompensation?
- Initially, the cardiovascular physiological changes during pregnancy had to be discussed (the order changed)
- Lines 224-225 How does premature birth in pregnant women with COVID-19 influence the perinatal outcome?
- Lines 225-226 Without a close connection to the subject
- Lines 227-230 Moved to section 3.3
- Line 266 Guo's study is unrelated to pregnant women with COVID-19. Discussions must be related to the topic.
- The way of birth is related to each cardiovascular disease. Specify if there are any changes
- Nothing was discussed about the severity of the COVID-19 infection and maternal complications.
- Discussions should be clear and concise
Kind regards
Author Response
Dear reviewers, we are so thankful for your helpful and valuable comments. The comments helped us improve our manuscript quality. In the revised version, we conducted a comprehensive, updated systematic search and updated our study. Also, as the included studies increased, we restructured and rewrote the results section and set new tables.
Reviewer 1:
- Line 29 myocardial infarction (MI) – why was the abbreviation made?
Thanks. We deleted the abbreviation. (page 1)
- Restructure the introduction
Thanks for your helpful comment. We edited and restructured the introduction thoroughly.
- Line 131 The mean age of cases was in the range of 26 to 45 – please specify the mean value
Thanks for your comment. We inserted the mean age. (highlighted in pink, page 4)
- Line 133-135 Please specify only cardiovascular comorbidities.
Thanks for your comment, edited. We deleted comorbidities that were not cardiovascular. (highlighted in turquoise, page 4)
- In table 1, maternal and fetal complications must be differentiated; to make possibly two tables
Thanks for your comment. We differentiated maternal and fetal cardiovascular complications into separate tables. In the revised version, we provided four tables: Table 1. characteristics and maternal outcomes of the included cohort/cross-sectional/case-control studies; Table 2. characteristics and fetal outcomes of the included cohort/cross-sectional/case-control studies; Table 3. characteristics and maternal outcomes of the included case report/case series studies; and Table 4. characteristics and fetal outcomes of the included case report/case series studies.
- Section 3.3 The fetal outcomes? It is an enumeration of some studies without a clear structuring of some ideas
Many thanks. We restructured this section. Also, we added the numbers and related p-values to make it more precise and clear.
- Section 3.4 Please restructure this section
Thanks for your comment. We tried to edit and restructure this section thoroughly.
- Section 3.7 The change of IR and IP at the level of the umbilical artery, what repercussions do they have at the level of the fetus? Correlation with maternal PI and RI at uterine level
Many thanks for your valuable comment. We added the data on the PI and RI at the uterine level reported by Anuk et al. to the results section (page 8) and table. (highlighted in turquoise)
- Nothing related to the physiopathological mechanisms of maternal and fetal cardiac decompensation?
Thanks. We added a paragraph explaining the physiopathological mechanisms of maternal and fetal cardiac decompensation. (highlighted in turquoise, pages 35 and 36)
- Initially, the cardiovascular physiological changes during pregnancy had to be discussed (the order changed)
Thanks. We added a paragraph on the cardiovascular physiological changes during pregnancy. Also, we changed the order as suggested. We inserted this paragraph as the second paragraph of the discussion before discussing COVID-19 complications. (highlighted in turquoise, page 35)
- Lines 224-225 How does premature birth in pregnant women with COVID-19 influence the perinatal outcome?
Thanks, we searched the literature thoroughly and as there were not enough data on how premature birth in pregnant cases can affect perinatal outcomes, we reported the results of previous meta-analysis studies on the association of COVID-19 during pregnancy and risk of premature birth or perinatal outcomes. (highlighted in turquoise, page 35)
- Lines 225-226 Without a close connection to the subject
Thanks. Deleted. (highlighted in turquoise, page 36)
- Lines 227-230 Moved to section 3.3
Thanks. We moved those sentences to section 3.3. (highlighted in turquoise, page 5 and 6)
- Line 266 Guo's study is unrelated to pregnantpre women with COVID-19. Discussions must be related to the topic.
Thanks. We deleted the explanation of that study. (highlighted in turquoise, page 36)
- The way of birth is related to each cardiovascular disease. Specify if there are any changes.
Many thanks for your comment. We reviewed the included articles and reported the data on the way of birth if mentioned. (highlighted in turquoise, in tables)
- Nothing was discussed about the severity of the COVID-19 infection and maternal complications.
Thanks. We added two paragraphs about the severity of the COVID-19 infection during pregnancy and maternal complications. (highlighted in turquoise, page 38)
- Discussions should be clear and concise
Thanks. We tried to edit the discussion and make it more clear. Also, we added the required information to make it more concise.
Reviewer 2 Report
The authors provide a systematic review on the maternal and fetal cardiovascular complication of COVID-19.
While the idea behind this review may be interesting (particularly, the topic), I unfortunately believe that this study have several limitations.
In detail:
- Abstract: please report number/measure of effect in the abstract (in this form, it is poorly informative of the actual results of the study).
- Introduction: "Also, myocardial injury and bradycardia are ex- 63
clusively explained in pregnant women with critical and severe COVID-19". This sentence is unclear - what does it means? Please revise.
- The aim of the study (which is introduced in the last three lines of introduction) is unclear and should be better defined: what is the primary aim of the study? summarise the current literature about what specific outcome/aspect of the relationship between COVID-19 and pregnancy? Next you can introduce the secondary aims, but specifiying those more clearly.
- Methods: was a protocol for this systematic review pre-registered on PROSPERO? If not, why?
- Methods: some more details on the methodology of this systematic review should be provided (i.e. how the screening was performed - please confirm that at least 2 authors (and specify who with the initials) independently and parallely assess the literature, etc.).
- Results: the table is not very readable in this form and it is very difficult to understand. Perhaps it would be better to resize the font and put this table in landscape. Please also be consistent regarding the decimal digits reported - also, when reporting percentages, it should be avoided to report in decimal form (.23) but instead they should be reported as numbers (23).
- Results: the most important criticism is that the data reported in the table are very non-specific and it is difficult to understand the main finding of each paper. For example, first row: "reported CV complication": preclamps". Description is a mere summary of the methods of this paper. It is impossible to understand what was the incidence of preeclampsia, if this was higher/lower than expected, etc. In short, this table is currently non useful and should be extensively reviewed to provide thorough insights of the studies included.
- Results: the previous criticism also regard the main manuscript text. There is barely no number reported for any of the outcomes investigated, but only aspecific statement ("reported that the incidence rate of 149
preeclampsia was higher among SARS-CoV-2 infected pregnant cases than pregnant cases without this infection." how much higher? What were the numbers? This type of reporting is not useful at all, does not give any information about the results of the study).
- Results: Another criticism is that reporting observation studies and case reports/case series together is confusing and do not give the right weight to different types of study (obviously, cohort studies are more important than case reports/case series).
- Another criticism is related to the fact that the findings reported are too vague in terms of definition of CVD. For example, the authors report about "arrhythmia", which is a very aspecific wording that comprise several type of arrhythmia, which incidence and relative influence of COVID-19 are a lot different. For example, AF has already been linked with COVID-19 (see for example a large meta-analysis 10.3390/jcm10112490 ), but other studies have reported about ventricular fibrillation, bradicardia etc. What are the differences between these entities? Similar remarks can be made for other disease (myocarditis, which is particularly important in this context; pulmonary hypertension/right ventricular dysfunction, which again was found associated with COVID-19 and is important in pregnancy). Currently this reviews fail to assess the contribution of these diseases.
- There is no discussion about the role of vaccination and COVID-19 treatments in regard to the incidence of CV complications during pregnancy.
Author Response
Dear reviewers, we are so thankful for your helpful and valuable comments. The comments helped us improve our manuscript quality. In the revised version, we conducted a comprehensive, updated systematic search and updated our study. Also, as the included studies increased, we restructured and rewrote the results section and set new tables.
Reviewer2:
- Abstract: please report number/measure of effect in the abstract (in this form, it is poorly informative of the actual results of the study).
Thanks, we added the maximum and minimum incidence of one maternal complication and one fetal complication. (Highlighted in yellow, page 1)
- Introduction: "Also, myocardial injury and bradycardia are exclusively explained in pregnant women with critical and severe COVID-19". This sentence is unclear - what does it means? Please revise.
Thanks. We revised this sentence. (highlighted in yellow, page 2)
- The aim of the study (which is introduced in the last three lines of introduction) is unclear and should be better defined: what is the primary aim of the study? summarise the current literature about what specific outcome/aspect of the relationship between COVID-19 and pregnancy? Next you can introduce the secondary aims, but specifiying those more clearly.
Thanks. We edited the paragraph representing aim of the study and tried to specify the aims more clearly. (highlighted in yellow, page 2)
- Methods: was a protocol for this systematic review pre-registered on PROSPERO? If not, why?
We conducted this systematic review according to PRISMA guidelines and Cochrane guidelines for systematic review studies. As PROSPERO registration and receiving its code takes time, we did not register. Whether you suggest we will register our systematic review on PROSPERO.
- Methods: some more details on the methodology of this systematic review should be provided (i.e. how the screening was performed - please confirm that at least 2 authors (and specify who with the initials) independently and parallely assess the literature, etc.).
Thanks. We added the requested information to the methods section and provided more details. (highlighted in yellow, page 2 and 3)
- Results: the table is not very readable in this form and it is very difficult to understand. Perhaps it would be better to resize the font and put this table in landscape. Please also be consistent regarding the decimal digits reported - also, when reporting percentages, it should be avoided to report in decimal form (.23) but instead they should be reported as numbers (23).
Thanks. We tried to resize the tables and put them in landscape. Also, we made the decimal digits consistent and edited the percentages.
- Results: the most important criticism is that the data reported in the table are very non-specific and it is difficult to understand the main finding of each paper. For example, first row: "reported CV complication": preclamps". Description is a mere summary of the methods of this paper. It is impossible to understand what was the incidence of preeclampsia, if this was higher/lower than expected, etc. In short, this table is currently non useful and should be extensively reviewed to provide thorough insights of the studies included.
Thanks for your valuable comment. We revised the table and inserted the incidence of complications reported. We provided four tables showing cardiovascular comorbidities and complications, separated according to maternal or fetal complications, and cohorts/ case-controls/ cross-sectional studies or case reports/ case series. We omitted the description column and reorganized the tables.
- Results: the previous criticism also regard the main manuscript text. There is barely no number reported for any of the outcomes investigated, but only aspecific statement ("reported that the incidence rate of preeclampsia was higher among SARS-CoV-2 infected pregnant cases than pregnant cases without this infection." how much higher? What were the numbers? This type of reporting is not useful at all, does not give any information about the results of the study).
We edited the mentioned issue in our text precisely. We reported the related numbers on prevalence of complications or p-values (if mentioned) in comparisons. The results section is edited extensively according to the comments. (page 3-8)
- Results: Another criticism is that reporting observation studies and case reports/case series together is confusing and do not give the right weight to different types of study (obviously, cohort studies are more important than case reports/case series).
Thanks. We edited the results section and tables according to this comment. In the results section’s test, we attempted to report cohorts or cross-sectional studies or case-controls initially in each sub-section and subsequently report the case reports- case series separately. Also, we provided different and separated tables for case reports/ case series.
- Another criticism is related to the fact that the findings reported are too vague in terms of definition of CVD. For example, the authors report about "arrhythmia", which is a very aspecific wording that comprise several type of arrhythmia, which incidence and relative influence of COVID-19 are a lot different. For example, AF has already been linked with COVID-19 (see for example a large meta-analysis 10.3390/jcm10112490 ), but other studies have reported about ventricular fibrillation, bradicardia etc. What are the differences between these entities? Similar remarks can be made for other disease (myocarditis, which is particularly important in this context; pulmonary hypertension/right ventricular dysfunction, which again was found associated with COVID-19 and is important in pregnancy). Currently this reviews fail to assess the contribution of these diseases.
Thanks for your valuable comment. In the revised version, we tried to explain the reported complications in more detail. However, the type of arrhythmias was not mentioned in most of the included studies. Also, we added detailed information to tables whether reported. We thoroughly investigated the mentioned meta-analysis study, but there were no data regarding pregnancy. We mentioned the issue of insufficient details on complications reported by included studies in the limitations section at the end of the discussion.
- There is no discussion about the role of vaccination and COVID-19 treatments in regard to the incidence of CV complications during pregnancy.
Thanks. According to the literature, we added information on the roles of vaccination and COVID-19 treatments, and cardiovascular complications during pregnancy. (highlighted in yellow, pages 38 and 39).